# Reliability Analysis of Serviceability Limit State for Braced Excavation Considering Multiple Failure Modes in Spatially Variable Soil

**Li Hong [1,2], Longlong Chen [1] and Xiangyu Wang [1,2,3,*]**

1   School of Civil Engineering, Chongqing University, Chongqing 400045, China;
    201816021017@cqu.edu.cn (L.H.); chenlonglong@cqu.edu.cn (L.C.)
2   Institute for Smart City of Chongqing University in Liyang, Chongqing University, Changzhou 213300, China
3   School of Design and the Built Environment, Curtin University, Bentley, WA 6102, Australia
*   Correspondence: xiangyu.wang@curtin.edu.au

**Abstract:** High uncertainty is an inherent behavior of geotechnical materials. Nowadays, random field theory is an advanced method to quantify the effect of high uncertainty on geotechnical engineering. This study investigates the effect of spatial variable soil layers on deformations of deep excavation via the random finite element method. A procedure based on PLAXIS 2D software was developed to generate two-dimension random finite element models including multiple variables. Via the K-S test and S-W test, the excavation deformations basically followed lognormal distribution. With the growth of standard deviation of soil properties parameters, the distribution of excavation deformations becomes wider, and the failure probability increases. When the vertical scale of fluctuation ranges from 1 m to 25 m, the distribution of excavation deformations becomes wider. To analyze system reliability, this study proposed a fitted multiple lognormal distribution methods, which was a method with higher efficiency. The results indicated that system reliability was lower than single failure probability and sensitive to design level. The system failure probability will be over-evaluated or under-evaluated if the correlation between excavation responses is ignored. This study provided a novel method to quantify the effect of high uncertainty of soil layer on excavation responses and proposed an efficient method for system reliability analysis, which is meaningful for excavation reliability design.

**Keywords:** spatial variability; system reliability; serviceability limit state; Cholesky decomposition; multiple log-normal distribution

## 1. Introduction

Braced excavations are widely used in underground structures and subway stations, which is an important part of the city building system and traffic system. There are two main requirements in reliability analysis, one is the requirement of ultimate limit state (ULS), and another is the requirement of serviceability limit state. For braced excavation, ULS requirements mainly include bearing capacity of support system, safety factor against basal heave and slide, etc. SLS requirements include limitation of wall deflection, ground settlement, basal heave, and so on. For most braced excavations in busy cities, in order to protect the serviceability of underground structures and surrounding tunnels, SLS requirements are usually stricter than ULS requirements. However, the deformations of excavation depend on many points. The uncertainty of parameters of soil property makes a great influence on evaluating deformations of excavation. Different from materials of structure engineering, the uncertainty is an inherent behavior of soil due to its geological history [1,2]. Vanmarcke and his colleagues pointed out that the variance of the spatially averaged soil property over some domains is less than the variance at the point [3]. The variance of soil property will decrease with the enlarging of the sizes of the domain.

Therefore, Vammarcke proposed the variance reduction function method. In addition, the random field theory is another effective method to address this problem [3]. The random finite element method (RFEM) is a combination of random field theory and finite element analysis, which can evaluate the effect of spatial variability on geotechnical engineering. There are three main methods to establish a random finite element model, including spatial averaging [4], Cholesky decomposition [5], and K-L expansion [6]. Random field theory has been applied in bearing capacity and settlement of shadow foundation [7,8], slope reliability [9–18], tunnel reliability [5,19], and so on.

For braced excavation, RFEM was adopted to analyze the effect of soil spatial variability on safety factors against basal heave [4,20]. Luo et al. evaluated the effect of soil vertical spatial variability on wall deflection, ground settlement, and structure responses of braced excavation [21]. Gong et al. proposed a new framework used for probabilistic analysis of the performance of a supported excavation [22]. Gholampour et al. applied conditional random field in reliability analysis of braced excavation in unsaturated soils [23]. On the other hand, in addition to spatial variability of soil properties, the correlation between multiple failure modes is also a key point to assess the serviceability reliability of braced excavation. In previous decades, the empirical method [24], field observation [25], statistical analysis [26], and machine learning method [27–29] was applied in the evaluation of deformations of braced excavation. Nowadays, more machining learning methods were applied in reliability analysis and risk assessment of deep excavation, such as extreme gradient boosting [30], optimization algorithms [31–33], support vector machine [34], and random-set finite element method [35]. However, the study about the effect of soil spatial variability on braced excavation was less reported. The system reliability analysis of deep excavation considering spatial variable soil layers and multiple failure modes is still a challenge. This study aims to explore the character of deformations induced by excavation in spatial variable soil layers and develop a system reliability model which can consider multiple failure modes.

In this study, random finite element models were automatically generated by MATLAB and PLAXIS 2D software. Via probabilistic analysis and parametric study, the effect of soil spatial variability on excavation responses was systematically investigated. Additionally, the authors analyzed the correlation of different failure modes in spatial variable soil layers. A system reliability model based on SLS was presented to evaluate system failure probabilities considering multiple failure modes.

## 2. Methodology

### 2.1. Random Finite Element Analysis Method

For RFEM, there are four main steps to generate random fields, concluding generation of random samples in standard normal space, determination of autocorrelation matrix $\rho$ and correlation matrix $R$, Cholesky decomposition, and mapping random fields.

For reliability analysis, failure samples were important, however, most failure samples were located in the tail of the distribution. Compared with other sampling methods, the Latin hypercube sampling technique covers the upper and lower bounds value of probability distribution more uniformly. Therefore, the initial random samples $\xi_{(n \times m)} = \left[ \xi^1_{(n \times 1)}, \xi^2_{(n \times 1)}, \cdots, \xi^m_{(n \times 1)} \right]$ are generated in standard normal space by a Latin Hypercube sampling technique. In general, five general types of autocorrelation functions were adopted to represent the autocorrelation of soil parameters in spatial space [36]. In this study, the exponential function was used, which could be expressed as Equation (1).

$$\rho\left(\tau_x, \tau_y\right) = \exp\left[ -2\left( \frac{\tau_x}{\delta_h} + \frac{\tau_y}{\delta_v} \right) \right] \tag{1}$$

where $\tau_x = \left| x_i - x_j \right|$, $\tau_y = \left| y_i - y_j \right|$, $(x_i, y_i)$ is the central coordinate $i$th of the random field element. $\delta_h$ and $\delta_v$ are the horizontal and vertical scales of fluctuation, respectively.

On the other hand, there was more than one random field (such as cohesion and friction angle), and the correlation matrix (*R*) was used to represent the relationship between different random fields in standard normal space. Furthermore, via the Cholesky decomposition of matrix $\rho$ and *R*, the Upper triangular matrix $L_1$ and $L_2$ can be obtained (Equation (2)).

$$\begin{cases} L_1 L_1^T = R_{(m \times m)} \\ L_2 L_2^T = \rho_{(n \times n)} \end{cases} \tag{2}$$

where *m* is the number of random fields, and *n* is the number of random finite elements.

Then, based on the Cholesky decomposition, the normal random fields were obtained by Equation (3):

$$G_{(n \times m)} = L_2 \cdot \xi_{(n \times m)} \cdot L_1 \tag{3}$$

Lastly, via the transformation equations between normal distribution and lognormal distribution (Equation (4)), the exponential random field can be determined:

$$\begin{cases} \sigma_{i_{\ln}} = \sqrt{\ln\left(1 + \sigma_i^2 / \mu_i^2\right)} \\ \mu_{i_{\ln}} = \ln \mu_i - 0.5\sigma_{i_{\ln}}^2 \\ E_{(n \times 1)}^i = \exp\left(\mu_{i_{\ln}} + \sigma_{i_{\ln}} \cdot G_{(n \times 1)}^i\right) \end{cases} \tag{4}$$

where $\mu_i$ and $\sigma_i$ are the mean value and standard deviation of *i*th random variables, $\mu_{i_{\ln}}$ and $\sigma_{i_{\ln}}$ are the corresponding value in lognormal space. $E_{(n \times 1)}^i$ and $G_{(n \times 1)}^i$ are the part of $E_{(n \times m)}$ and $G_{(n \times m)}$, respectively. $\left(E_{(n \times m)} = \left(E_{(n \times 1)}^1, E_{(n \times 1)}^2, \cdots E_{(n \times 1)}^m\right), G_{(n \times m)} = \left(G_{(n \times 1)}^1, G_{(n \times 1)}^2, \cdots, G_{(n \times 1)}^m\right)\right)$.

*2.2. Deterministic Finite Element Model*

In this study, the finite element software PLAXIS 2D was adopted to model braced excavations. The numerical case was referred in previous studies by Goh et al. [4]. The value of *m* ranges from 0 to 1, which decreases with the growth of soil stiffness [37]. Figure 1 shows the deterministic analysis finite element model (FEM). Due to the symmetry of the model, only the left half of the excavation was modeled. The horizontal length and vertical height of the model were set at 70 m and 60 m, respectively. The groundwater table is located 2.0 m below the ground surface. In the FEM simulation, the hardening soil model (HS) and hardening soil with a small strain model (HSS) are used to simulate deep excavation. However, in this study, clay soil was considered a spatial variable layer. For HS and HSS models, it is hard to evaluate the spatial variability in random finite element analysis. As recommended by Luo et al. [4,21], the Mohr-Coulomb model (MC) and HS model were used to simulate the clay soil layer and sand soil layer, respectively. The drained type, and undrained type A was adopted to simulate the sand layer and clay layer, respectively. The struts were modeled as spring elements, and the diaphragm walls were modeled as linear elastic materials. Detailed construction steps and parameters for retaining the structural system were shown in Table 1. The construction steps of excavations followed the principle of dewatering water first, installing struts later, and excavating finally. The depth of each excavation stage was 4 m. The depth of four struts is 1 m, 5 m, 9 m, and 13 m, respectively. For random finite element models, their calculation steps were the same as the deterministic model. Because of larger samples of random analysis, to improve calculation efficiency, the installation of struts and excavation were coupled into one construction step. The parameters for soil properties were shown in Table 2.

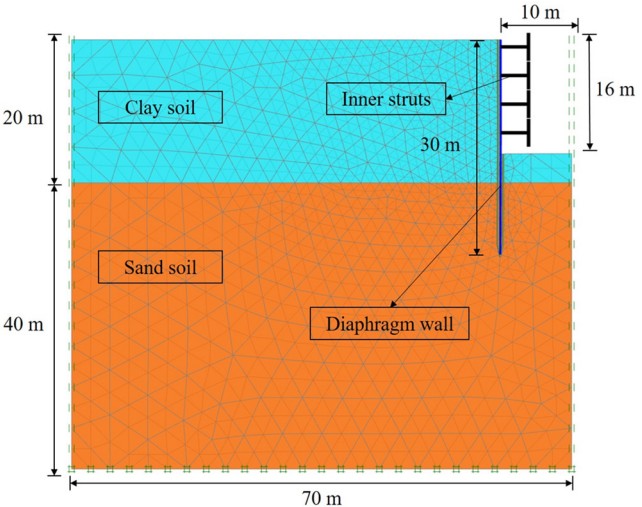

**Figure 1.** Finite element model for deterministic analysis.

**Table 1.** Construction steps and parameters of retaining structural system.

| Stage | Construction | Parameter of Retaining Structural System | |
|---|---|---|---|
| | | *EA* (kN) | *EI* (kN·m$^2$) |
| 1 | Initial confined consolidation | - | - |
| 2 | Activate diaphragm wall | $3 \times 10^6$ | $2.5 \times 10^6$ |
| 3 | Lower the ground water to GL −9 m | - | - |
| 4 | Excavate to −4 m and activate the first strut at GL −1 m | $2 \times 10^6$ | - |
| 5 | Excavate to −8 m and activate the second strut at GL −5 m | $2 \times 10^6$ | - |
| 6 | Lower the ground water to GL −17 m | - | - |
| 7 | Excavation to GL −12 m and activate the third strut at GL −9 m | $2 \times 10^6$ | - |
| 8 | Excavate to GL −16 m and activate the fourth strut at GL −13 m | $2 \times 10^6$ | - |

Notes: GL is ground level in finite element model.

**Table 2.** Parameters of soil properties.

| Number | Soil Layer | *T* | $\gamma$ | $\nu$ | *E* | $c_u$ | $\varphi$ | Ψ | $E_{50}^{ref}$ | $E_{oed}^{ref}$ | $E_{ur}^{ref}$ | *m* |
|---|---|---|---|---|---|---|---|---|---|---|---|---|
| | | (m) | (kN/m$^3$) | - | (MPa) | (kPa) | (°) | (°) | (MPa) | (MPa) | (MPa) | - |
| 1 | Clay | 20 | 17 | 0.3 | 9 | 30 | 20 | - | - | - | - | - |
| 2 | Sand | 40 | 19 | - | | 0 | 32 | 2 | 40 | 40 | 120 | 0.5 |

Notes: *T* is thickness of each soil layer. $\gamma$ is unit weight of soil layer. *E*, $c_u$, $\varphi$ is elastic modulus, soil cohesion and friction angle for MC model.

The horizontal distance of struts equals to 5 m.

$\psi$ is dilation angle. $\psi = \varphi - 30°$; if $\varphi < 30°$, $\psi = 0°$. From PLAXIS manual [37].

$E_{50}^{ref}$, $E_{oed}^{ref}$ and $E_{ur}^{ref}$ were reference secant stiffness in standard drained triaxial test, reference secant stiffness for primary oedometer loading, reference unloading/reloading stiffness, respectively. $E_{50}^{ref} : E_{oed}^{ref} : E_{ur}^{ref} = 1 : 1 : 3$, From PLAXIS manual [37].

*m* is power for stress-level dependency of stiffness. The value of m ranges from 0 to 1, which decreases with the growth of soil stiffness [37].

Figure 2 shows the ground settlement curve and wall deflection curve during deep excavation. The ground settlement curve was similar to "*V*" and the wall deflection curve was similar to a bow, which is the classical character of deep excavation with inner struts [38,39]. The maximum value of ground settlement equaled 2% of excavation depth, which was 0.7 times of excavation depth away from the excavation boundary. The major settlement region ranged from 0–2 times of excavation depth, and the secondary settlement

region ranged from 2–4 times of excavation depth. The wall deflection increases with the growth of excavation depth. The maximum wall deflection equaled 4.5% of excavation depth, which is located at the final depth of excavation.

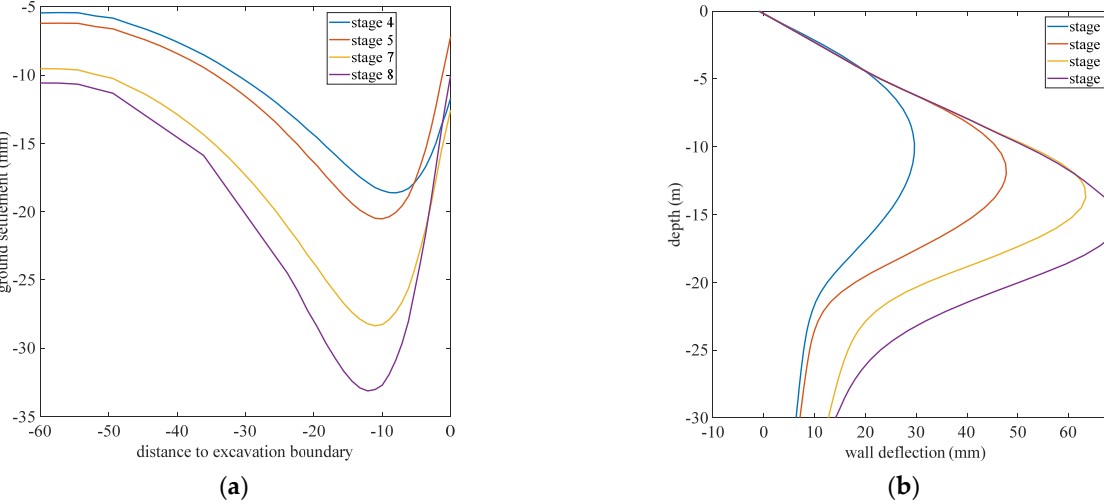

(a)                     (b)

**Figure 2.** Deformation curve of deterministic model: (**a**) ground settlement curve; (**b**) wall deflection curve.

### 2.3. Modeling of Braced Excavation in Spatial Variability Soil

Based on the deterministic finite element model, the two key parameters (cohesion $c_u$ and friction angle $\varphi$) of clay soil were considered, as two negative correlated random fields, the relationship factor between $c_u$ and $\varphi$ was equal to −0.5 [9]. For clay soil, there is a high linear correlation between elastic modulus and cohesion of soil [4], which can be approximately represented by empirical formulas. In addition, the high correlation between two random fields would disturb the Cholesky decomposition, it is hard to calculate Equation (2). Therefore, it is necessary to separate random variables with high correlation from the correlation matrix. Consequently, in this study, elastic modulus (*E*) is set as $E = 300c_u$ [4,40]. In other words, *E* is considered a random field that is related to $c_u$. Other parameters are treated as constant.

In a random finite element model, spatial variability of soil was mapped onto each finite element mesh. For PLAXIS 2D software, automation of random finite element modeling was conducted by the following Figure 3.

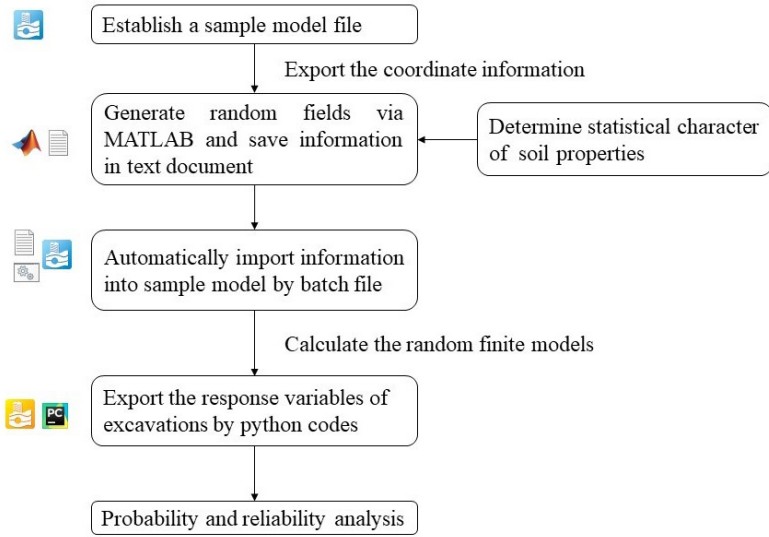

**Figure 3.** Flow chart of random finite element method.

In Figure 3, there are five steps:

(1) Establish a sample model of the random finite element method. Different from the deterministic model shown in Figure 1, the clay layer was divided into 560 regions, corresponding to random field elements. Meanwhile, the statistical characteristics of random fields are determined, such as the mean value ($\mu$), Coefficient of variation (*COV*), correlation matrix *R*, and fluctuation of scale in horizontal and vertical directions ($\delta_x$ and $\delta_y$).

(2) Based on the statistical characters in step 1, generate two negative correlated random fields of $c_u$ and $\varphi$ via MATLAB codes, from Equations (1)–(4). The random fields of $c_u$ or $\varphi$ can be expressed as strength cloud charts (Figure 4).

(3) Via batch file and command codes in PLAXIS 2D, automatically import the information of random fields into finite element mesh. Similarly, calculate the random finite element models automatically by batch files and PLAXIS codes.

(4) Export response data of excavations into EXCEL by Python codes [37], such as wall deflection, ground settlement, basal heave, and bending moment of the diaphragm wall.

(5) Using the response data in step 5, calculate single mode failure probability and system failure probability.

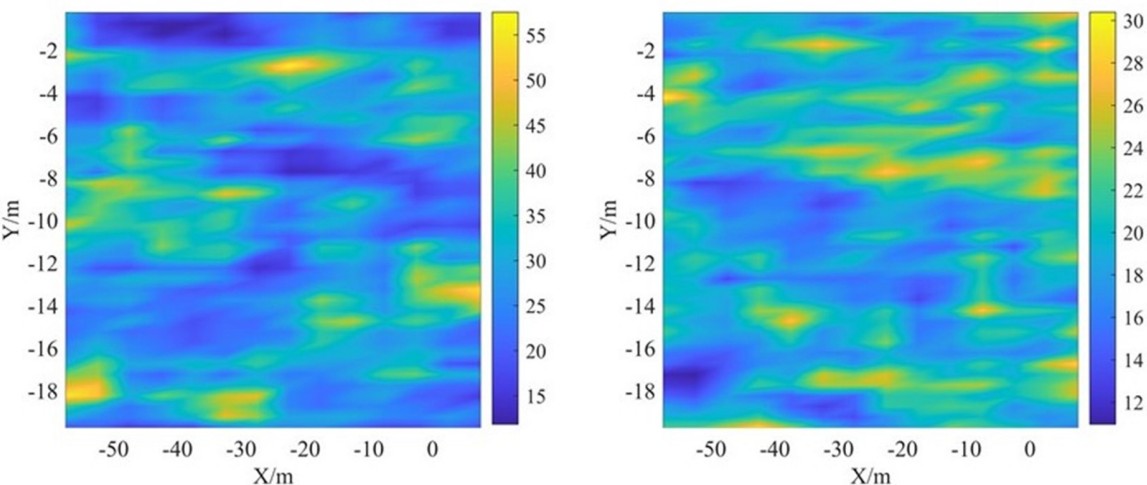

**Figure 4.** Random field realizations of $c_u$ (**left**) and $\varphi$ (**right**) (case 3).

From Figure 4, the value of $c_u$ and $\varphi$ obey the three-deviation criterion. With the increase of fluctuation of scale, the deviation of $c_u$ or $\varphi$ will decrease, and the random finite element models gradually transform into random variable models.

From statistics from Phoon et al. [1,2], the uncertainty of friction angle is lower than soil cohesion or undrained shear strength. Therefore, there are 14 cases in this paper, as shown in Table 3. Three levels of cohesion ($COVc_u$ = 10%, 20%, 30%), three levels of friction angle ($COV\varphi$ = 10%, 15%, 20%), six levels of vertical distance $\delta_y$ = 1 m, 2.5 m, 5 m, 10 m, 25 m, 50 m are considered. Cases 1~9 aimed to evaluate the effect of spatial variability of soil properties on responses of excavation. Cases 3, 10~14 aimed to evaluate the effect of the scale of fluctuation on responses.

The number of Latin hypercube sampling in each case is 500, which can guarantee the convergence of calculation results (Figure 5).

**Table 3.** Values of parameters in design cases.

| Case Number | $COVc_u$ | $\sigma_{c_u}$/kPa | $COV\varphi$ | $\sigma_\varphi$/° | $\delta_y$ (m) | $\delta_x$ (m) |
|---|---|---|---|---|---|---|
| 1 | 30% | 9 | 10% | 2 | 2.5 | 25 |
| 2 | 30% | 9 | 15% | 3 | 2.5 | 25 |
| 3 | 30% | 9 | 20% | 4 | 2.5 | 25 |
| 4 | 20% | 6 | 10% | 2 | 2.5 | 25 |
| 5 | 20% | 6 | 15% | 3 | 2.5 | 25 |
| 6 | 20% | 6 | 20% | 4 | 2.5 | 25 |
| 7 | 10% | 3 | 10% | 2 | 2.5 | 25 |
| 8 | 10% | 3 | 15% | 3 | 2.5 | 25 |
| 9 | 10% | 3 | 20% | 4 | 2.5 | 25 |
| 10 | 30% | 9 | 20% | 4 | 1 | 25 |
| 11 | 30% | 9 | 20% | 4 | 5 | 25 |
| 12 | 30% | 9 | 20% | 4 | 10 | 25 |
| 13 | 30% | 9 | 20% | 4 | 25 | 25 |
| 14 | 30% | 9 | 20% | 4 | 50 | 25 |

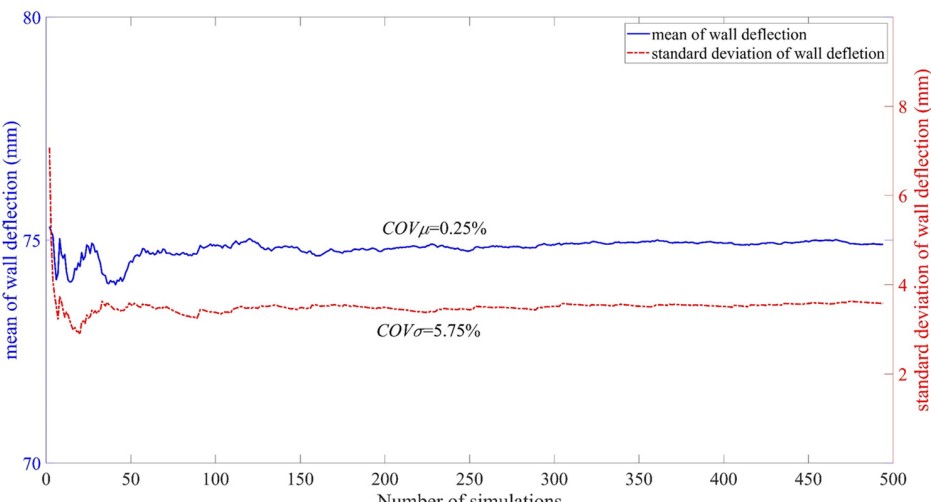

**Figure 5.** Convergence curve of wall deflection (case 1).

Taken case 1 for example, as shown in Figure 5, $COV\mu$ and $COV\sigma$ of wall deflection are 0.25% and 5.75% when the number of simulations equals to 500, which is satisfied with the condition of convergence [5,41].

## 3. Effect of Spatial Variability on Responses of Braced Excavation

### 3.1. The Influence of $COVc_u$ and $COV\varphi$

In this part, the effect of spatial variability on responses of excavation (wall deflection, ground settlement, and basal heave, etc.) is investigated. Following the automatic RFEM procedure developed in this study (Figure 3), the mean value and standard deviation of responses of excavation in each case are shown in Appendix A.

The statistical characters of responses to excavation are shown in Figure 6. For example, from Figure 6a, the histogram of wall deflection in case 3 is similar to log-normal distribution, and the corresponding fit curve of the log-normal probability density function is displayed by a black dash-dotted line. In order to validate the assumption, the goodness of fit test is a useful method. Via Kolmogorov-Smirnov (KS) test with a 95% confidence interval, the *p* value of the KS-test was 0.9905. Via Shapiro-Wilk (SW) test with a 95% confidence interval, the *p* value of the SW-test was 0.878 and the value of test statistics was 0.998. Both KS-test and SW-test supported that wall deflection followed log-normal distri-

bution. In addition, in the Quantile-Quantile plot of wall deflection shown in Figure 6b, the quantiles of lognormal distribution are basically consistent with the quantiles of samples.

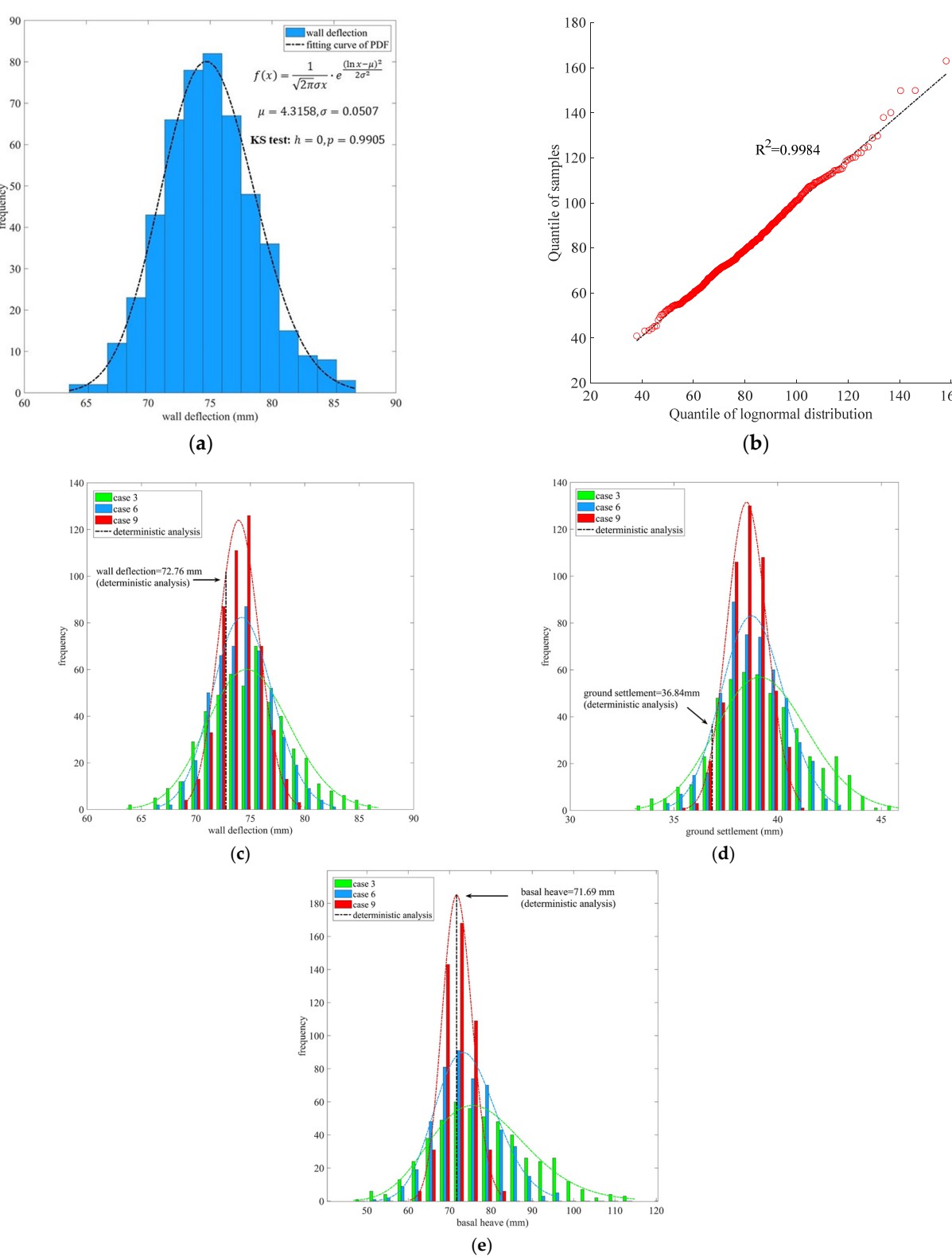

**Figure 6.** Statistical distribution of excavation responses: (**a**) histogram of wall deflection in case 3; (**b**) Quantile-Quantile plot of wall deflection; (**c**) histogram of wall deflection with different $COVc_u$; (**d**) histogram of ground settlement with different $COVc_u$; (**e**) histogram of basal heave with different $COVc_u$.

The coefficient of determination is 0.9984, which also verifies the reasonableness of the assumption. The other responses of excavation in each case are shown in Appendix A, most of them can be considered as random variables following log-normal distribution. By considering responses of excavation as log-normal distribution models, single-mode failure probability and system mode failure probability can be evaluated, this part will be discussed in Section 4.

As shown in Figure 6c–e, the mean value of basal heave is close to the value in the deterministic model. Different from the mean value of basal heave, the mean value of ground settlement and wall deflection is a little higher than the value of the deterministic model. Based on random finite element analysis for foundation capacity, Griffiths proposed that ground settlement was subjective to the influence of lower stiffness soil [8]. Furthermore, with the increase of $COVc_u$, the $COV$ of responses of excavation increased. For example, in case 3 ($COVc_u$ = 30%, $COV\varphi$ = 20%), the range of basal heave was 46–114 mm, while in case 9 ($COVc_u$ = 10%, $COV\varphi$ = 20%), the range of basal heave was 53–96 mm. It is noticeable that the COV of basal heave was 15.657% in case 1, while the $COV$ of ground settlement and wall deflection was 6.644% and 4.778%, respectively. Basal heave was more easily affected by spatial variability of soil properties.

Furthermore, the influence of $COV\varphi$ and $COVc_u$ on the responses of excavation is shown in a boxplot chart. In Figure 7, the boxplot is divided into three parts, which represent $COVc_u$ = 10%, 20%, and 30%, respectively. Each part contains three cases, representing $COV\varphi$ = 10%, 15%, 20%. In each boxplot, the black is the mean value of each case, which is close to the median values. It is proved that the excavation responses approximately follow a log-normal distribution. Compared with soil cohesion ($c_u$) and elastic modulus ($E$), friction angle ($\varphi$) makes a lower influence on responses of excavation. There are few changes for median value with different $COV\varphi$, only the standard deviation slightly increases with the growth of $COV\varphi$. The mean value of wall deflection and basal heave slightly increases with the growth of $COVc_u$. However, $COVc_u$ has no effect on the mean value of the ground settlement.

Spatial variability also makes a great influence on the failure path, for example, in the 500 models in case 3, the maximum value of basal heave is 114.8 mm, while the minimum value of basal heave is 46.7 mm. The strain cloud chart of these two models is shown in Figure 8. In the first model, the main deformation is basal heave, the second deformation is a deflection of the wall. On the contrary, in the second model, wall deflection is the main deformation.

**Figure 7.** The effect of $COVc_u$ and $COV\varphi$ on excavation responses: (**a**) Boxplot of wall deflection; (**b**) Boxplot of ground settlement; (**c**) Boxplot of basal heave.

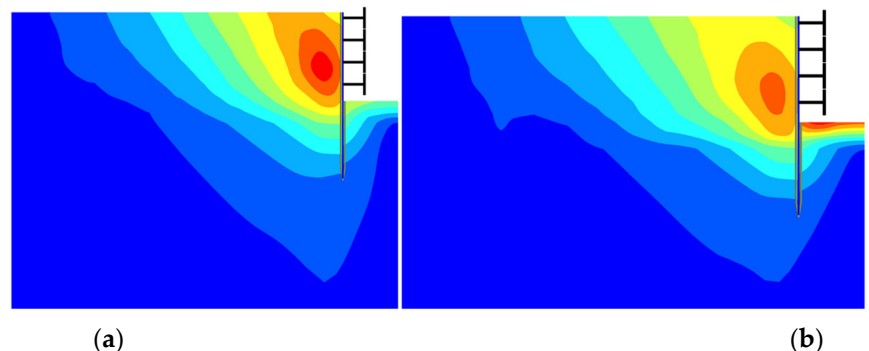

(**a**)                       (**b**)

**Figure 8.** Effect of spatial variability on failure mode: (**a**) basal heave dominate; (**b**) wall deflection dominate.

### 3.2. The Influence of Vertical Fluctuation of Scale

From Table 3, case 3 and cases 10~14 are used to investigate the effect of vertical fluctuation of scale ($\delta_v$) on excavation responses. Similarly, the curves of the fitting probability density function of each case are shown in Figure 9. The distribution is wider with an

increase of $\delta_y$ when $\delta_y$ ranges from 1 m~25 m. When $\delta_y$ is larger than 25 m, the distributions of excavation responses basically remain convergence. Compared with $COVc_u$ and $COV\varphi$, $\delta_y$ has a larger impact on excavation responses. Based on the statistical data, $\delta_y$ of soil usually ranges from 1 m~6 m, and the excavation depth is usually at the level of 5 m~50 m. Therefore, it is necessary to determine the value of $\delta_y$ via field observation data and geological statistic method.

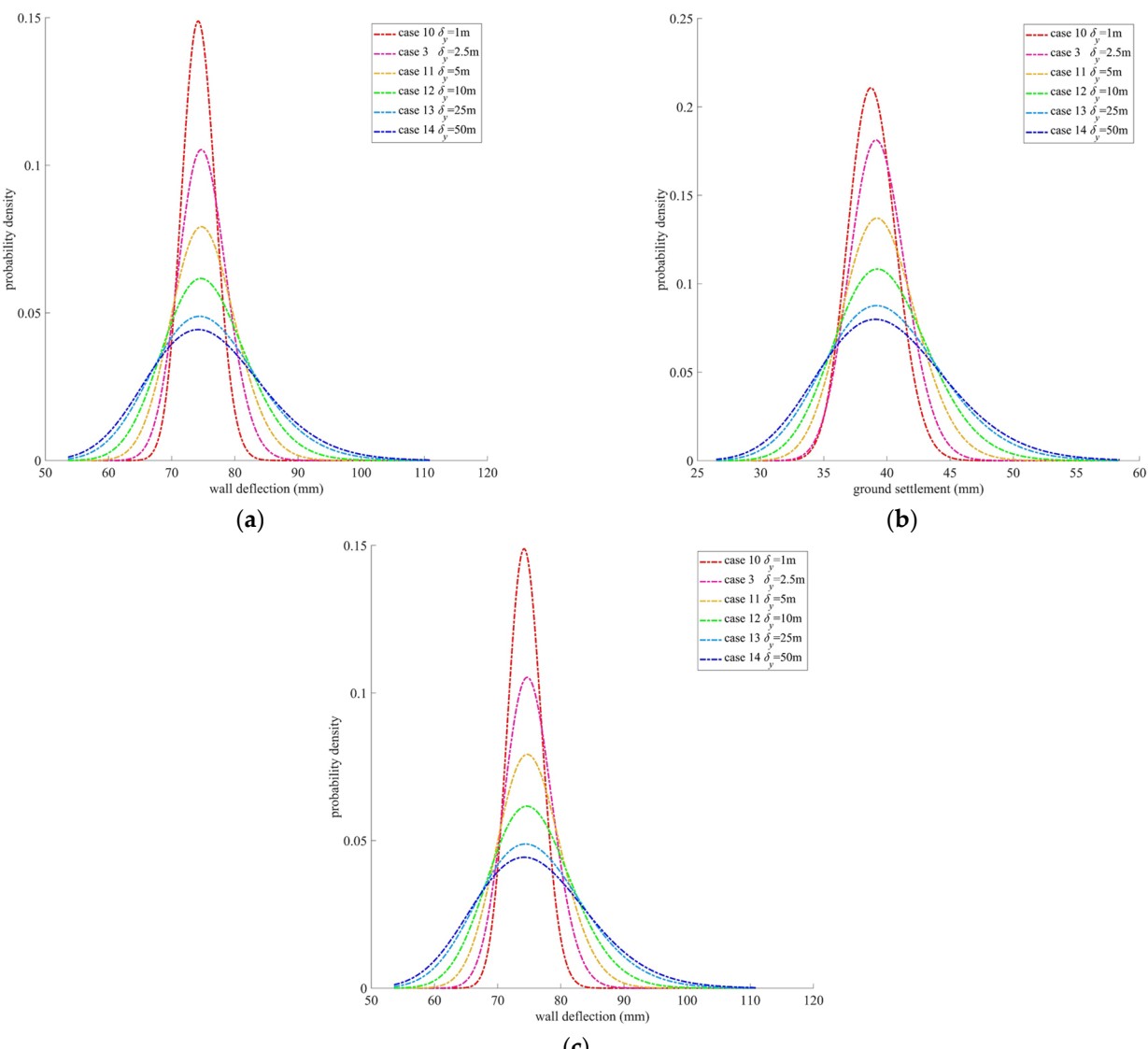

**Figure 9.** Effect of $\delta_y$ on distribution of excavation responses: (**a**) histogram of wall deflection with different $\delta_y$; (**b**) histogram of ground settlement with different $\delta_y$; (**c**) histogram of basal heave with different $\delta_y$.

### 3.3. Correlation Analysis of Excavation Responses

Based on the results of random finite element models, the relationship between responses of excavation can be described by the Spearman correlation matrix. Different from the Pearson matrix, the Spearman matrix can describe a nonlinear relationship between variables. Spearman correlation matrix in Case 3 is shown in Figure 10.

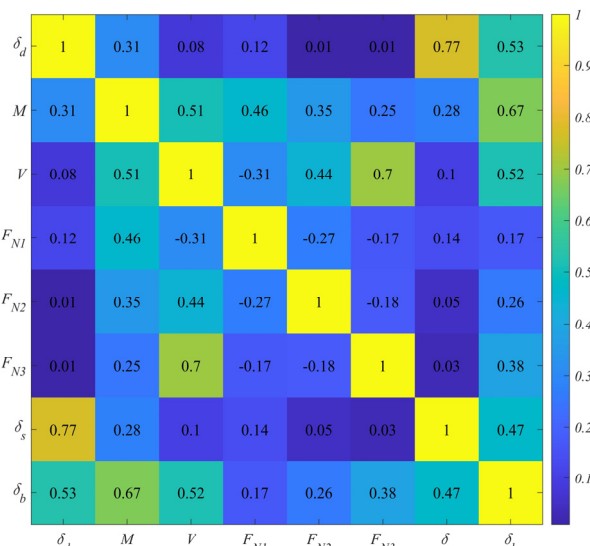

**Figure 10.** Spearman correlation matrixes of responses (Case 3). Notes: $\delta_d$, $\delta_s$, $\delta_b$ are the vector of wall deflection, ground settlement and basal heave, respectively. *M*, *V* are the vector of bending moment and shear force of diaphragm wall, respectively. $F_{Ni}$ is the vector of axial force of strut in *i*th layer.

The relationship between excavation responses in the deterministic model was reported by some previous studies [26]. In the probabilistic model, there are also some similar tendencies. From Figure 10, there is a high level of linear correlation between $\delta_d$ and $\delta_s$ (0.77 for Spearman matrix). In addition, there is also a middle level of correlation between $\delta_d$ and $\delta_b$, $\delta_s$ and $\delta_b$, *M* and *V* (0.53, 0.47, 0.51 for Spearman matrix). The axial force of struts has a low level of correlation with other responses. However, the correlation between responses of excavation plays an important role in system reliability analysis, which is introduced in Section 4 in detail.

## 4. System Reliability Model Based on Serviceability Limit State

Generally speaking, most deep excavations are designed as underground structures, such as subway stations, underground commercial streets, and parking lots. Therefore, in order to guarantee the safety and serviceability of underground structures, the excavations should satisfy the demand of capacity and deformation at the same time, which is called the ultimate limit state (ULS) and serviceability limit state (SLS), respectively. In this study, a system reliability model based on the serviceability limit state was proposed. The serviceability limit state function of excavations can be expressed as follows:

$$\begin{cases} f(\delta_d) = \delta_d - \delta_{d\max} \\ f(\delta_s) = \delta_s - \delta_{s\max} \\ f(\delta_b) = \delta_b - \delta_{b\max} \end{cases} \tag{5}$$

where $\delta_d, \delta_s, \delta_b$ is wall deflection, ground settlement, and basal heave, respectively, which are considered as random variables following log-normal distribution. The key parameters of the log-normal distribution are obtained from Section 3.1. $\begin{bmatrix} \delta_{d\max} & \delta_{s\max} & \delta_{b\max} \end{bmatrix}^T$ is the limit deformations of excavations based on the standards, rules, or empirical formulas. In this study, the thresholds were obtained from Technical specifications for retaining and protecting building foundation excavations (JGJ 120-2012) [42], Code for Design of Build Foundation (GB 50007-2011) [43].

From Equation (5), the single failure mode probability can be expressed as follows:

$$\begin{cases} p_{f_1} = \int_{\delta_{d\max}}^{+\infty} f(\delta_d) d\delta_d \\ p_{f_2} = \int_{\delta_{s\max}}^{+\infty} f(\delta_s) d\delta_s \\ p_{f_3} = \int_{\delta_{b\max}}^{+\infty} f(\delta_b) d\delta_b \end{cases} \tag{6}$$

where $f(x_i)$ is the probability density function of log-normal distribution, $f(x_i) = \frac{1}{\sqrt{2\pi}\sigma_i x_i} \cdot \exp\left[\frac{-(\ln x_i - \mu_i)^2}{2\sigma_i^2}\right]$ $x_i \in (-\infty, +\infty)$.

Via numerical integral methods (in MATLAB 2020a software, the function name is *integral* and *logncdf*), the failure probabilities in Equation (6) were determined. For example, according to JGJ 120-2012, the limit wall deflection of excavations ($\delta_{d\max}$) depends on the design level, excavation depth (*H*), and surrounding building environment. In general, $\delta_{d\max}$ equals to 65 mm (0.4%*H*), 80 mm (0.5%*H*), 95 mm (0.6%*H*) for first, second and third level of excavations, respectively. The failure probabilities are 0.997, $9.54 \times 10^{-2}$, and 0 (the value is lower than $10^{-16}$ in MATLAB, which is considered 0). With different values of $\delta_{s\max}$, there is a great difference between failure probabilities, which is shown in Figures 11 and 12.

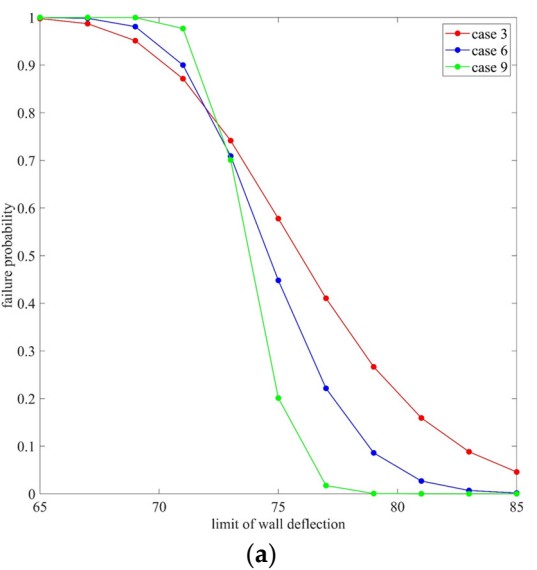
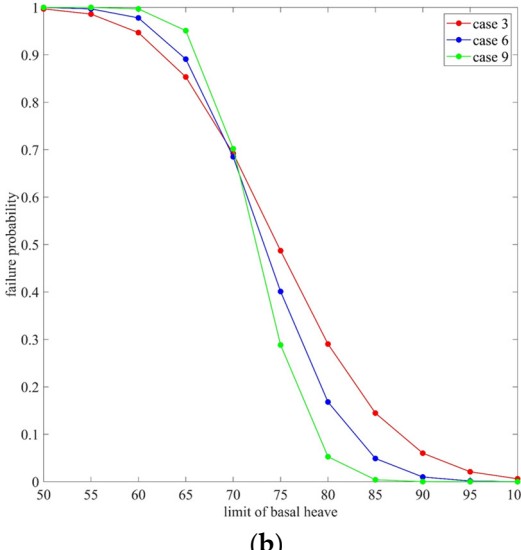

(**a**)          (**b**)

**Figure 11.** Spatial variability effects on failure probability: (**a**) different thresholds of wall deflection; (**b**) different thresholds of basal heave.

From Figure 11, $\delta_{s\max}$ ranges from 65 mm to 85 mm. The curves represent failure probability at different thresholds of wall deflection ($p_{f_1}$) with $COVc_u = 30\%, 20\%, 10\%$, respectively. With the increase of $COVc_u$, the gradient of the curve gradually decreases. For $COVc_u = 10\%$, the failure probability dramatically changes at 70~76 mm. However, for $COVc_u = 10\%$, the failure probability dramatically changes at 65~85 mm, which contains more uncertainty. In other words, for the same level of deformation thresholds, the larger $COVc_u$ is, the larger the failure probability is. Similar conclusions can be reached on basal heave failure mode from Figure 11b. Figure 12 shows the effect of $\delta_y$ on failure probability. The notable change of failure probability based on wall deflection limitation occurs when $\delta_b$ ranges from 55 mm to 105 mm, which is larger than $COVc_u$ (Figure 11a). At the same level of wall deflection limitation, the failure probability grows with $\delta_b$. The reason is that the fit distribution function becomes wider with the increase of $\delta_b$, which is mentioned in Section 3.2.

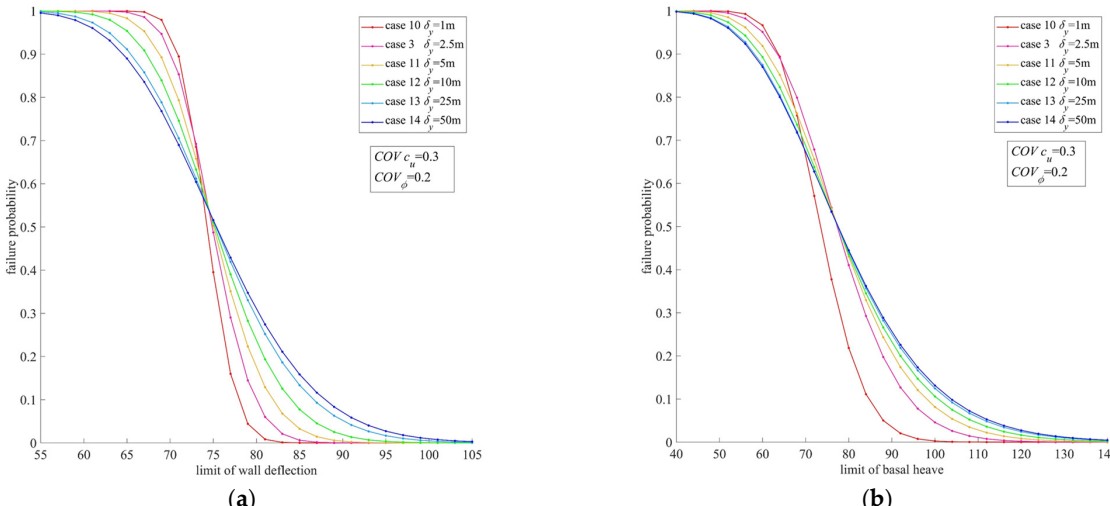

**Figure 12.** Effects of $\delta_y$ on failure probability at different thresholds of: (**a**) wall deflection; (**b**) basal heave.

Different from traditional reliability analysis, the system reliability model should take multiple failure modes into account. Every failure mode is considered as an element in an integral system, while multiple failure modes are considered as a series system, parallel system, or series-parallel system. In order to guarantee safety and serviceability, a series system was adopted in this study. Therefore, the system failure probability can be evaluated by a multidimensional lognormal distribution model:

$$p_f^{sys} = 1 - \int_{-\infty}^{\delta_{d\max}} \int_{-\infty}^{\delta_{s\max}} \int_{-\infty}^{\delta_{b\max}} f(\delta_d, \delta_s, \delta_b) d\delta_d d\delta_s d\delta_b \tag{7}$$

where $f(\delta_d, \delta_s, \delta_d)$ is the three-dimensional log-normal distribution model.

The three-dimensional log-normal distribution model can be expressed as Equation (8):

$$f(x_1, x_2, \cdots, x_n) = \frac{1}{(2\pi)^{n/2} |C|_{n \times n} \prod_{i=1}^{n} x_i} \exp\left[-\frac{1}{2}(\ln x - \mu)^T \cdot C \cdot (\ln x - \mu)\right] \tag{8}$$

where $x$ is the vector of random variables, $\mu$ is the vector of the mean value of random variables, $C$ is the covariance matrix of random variables.

From Equation (8), the correlation matrix is a key parameter to determine covariance matrix $C$. In this study, the Pearson matrix was adopted to calculate system failure probability. There are two main reasons:

(1) Compared with the Spearman correlation matrix, the Pearson matrix only describes the linear correlation of random variables. However, in the system reliability analysis of the underground pipe gallery, Fu et al. adopted the Pearson correlation factor to characterize the correlation between multiple failure modes [44,45].

(2) On the other hand, there is no doubt that the Pearson correlation factor is available for normal distribution. Other distributions can be transformed into standard normal space via Nataf transformations. One important assumption of Nataf transformation is that the correlation between random variables will not change during Nataf transformation. Moreover, Nataf transformation is only available for the Pearson correlation factor. Therefore, the Pearson correlation factor is still the main method to characterize the correlation of random variables in system reliability analysis.

Based on the above, the Pearson correlation matrix is used to establish a three-dimensional log-normal distribution model. Via the numerical integral method (in MAT-LAB 2020a software, the function called *integral3*), Equation (7) can be calculated easily. In addition, the function *mvncdf* in MATLAB2020a is also available if the three-dimensional

log-normal distribution model was transformed into a three-dimensional normal distribution model via Nataf transformation. From JGJ 120-2012 and GB 50007-2011 [42,43], the failure probability of single failure mode and system failure probability in each case are shown in Table 4.

**Table 4.** Probabilities of failure in different cases.

| Case Number | Design Level | $p_{f_1}$ | $p_{f_2}$ | $p_{f_3}$ | $p_f^{sys}$ |
|---|---|---|---|---|---|
| 1 | I | $9.983 \times 10^{-1}$ | $3.916 \times 10^{-1}$ | $9.467 \times 10^{-1}$ | $9.991 \times 10^{-1}$ |
| | II | $8.060 \times 10^{-2}$ | $1.787 \times 10^{-2}$ | $5.682 \times 10^{-1}$ | $5.757 \times 10^{-1}$ |
| | III | $2.874 \times 10^{-7}$ | $9.487 \times 10^{-5}$ | $1.571 \times 10^{-1}$ | $1.571 \times 10^{-1}$ |
| 2 | I | $9.981 \times 10^{-1}$ | $3.649 \times 10^{-1}$ | $9.490 \times 10^{-1}$ | $9.989 \times 10^{-1}$ |
| | II | $8.533 \times 10^{-2}$ | $6.780 \times 10^{-3}$ | $5.705 \times 10^{-1}$ | $5.771 \times 10^{-1}$ |
| | III | $4.475 \times 10^{-7}$ | $6.327 \times 10^{-6}$ | $1.555 \times 10^{-1}$ | $1.555 \times 10^{-1}$ |
| 3 | I | $9.974 \times 10^{-1}$ | $3.699 \times 10^{-1}$ | $9.513 \times 10^{-1}$ | $9.986 \times 10^{-1}$ |
| | II | $9.535 \times 10^{-2}$ | $7.510 \times 10^{-3}$ | $5.778 \times 10^{-1}$ | $5.857 \times 10^{-1}$ |
| | III | $1.26 \times 10^{-7}$ | $8.15 \times 10^{-6}$ | $1.592 \times 10^{-1}$ | $1.592 \times 10^{-1}$ |
| 4 | I | $1.000$ | $1.988 \times 10^{-1}$ | $9.779 \times 10^{-1}$ | $1.000$ |
| | II | $1.022 \times 10^{-2}$ | $3.134 \times 10^{-5}$ | $4.288 \times 10^{-1}$ | $4.294 \times 10^{-1}$ |
| | III | $1.343 \times 10^{-14}$ | $4.362 \times 10^{-12}$ | $2.441 \times 10^{-2}$ | $2.441 \times 10^{-2}$ |
| 5 | I | $1.000$ | $2.013 \times 10^{-1}$ | $9.791 \times 10^{-1}$ | $1.000$ |
| | II | $1.495 \times 10^{-2}$ | $3.537 \times 10^{-5}$ | $4.355 \times 10^{-1}$ | $4.366 \times 10^{-1}$ |
| | III | $3.890 \times 10^{-13}$ | $6.022 \times 10^{-12}$ | $2.511 \times 10^{-2}$ | $2.511 \times 10^{-2}$ |
| 6 | I | $9.999 \times 10^{-1}$ | $2.170 \times 10^{-1}$ | $9.809 \times 10^{-1}$ | $9.999 \times 10^{-1}$ |
| | II | $2.300 \times 10^{-2}$ | $6.758 \times 10^{-5}$ | $4.481 \times 10^{-1}$ | $4.501 \times 10^{-1}$ |
| | III | $1.538 \times 10^{-11}$ | $3.260 \times 10^{-11}$ | $2.676 \times 10^{-2}$ | $2.676 \times 10^{-2}$ |
| 7 | I | $1.000$ | $2.248 \times 10^{-2}$ | $9.998 \times 10^{-1}$ | $1.000$ |
| | II | $6.651 \times 10^{-6}$ | $3.331 \times 10^{-16}$ | $1.744 \times 10^{-1}$ | $1.744 \times 10^{-1}$ |
| | III | $0.000$ | $0.000$ | $2.576 \times 10^{-6}$ | $2.576 \times 10^{-1}$ |
| 8 | I | $1.000$ | $3.731 \times 10^{-2}$ | $9.998 \times 10^{-1}$ | $1.000$ |
| | II | $1.037 \times 10^{-4}$ | $1.922 \times 10^{-13}$ | $1.847 \times 10^{-1}$ | $1.847 \times 10^{-1}$ |
| | III | $0.000$ | $0.000$ | $3.105 \times 10^{-1}$ | $3.105 \times 10^{-1}$ |
| 9 | I | $1.000$ | $6.539 \times 10^{-2}$ | $9.998 \times 10^{-1}$ | $1.000$ |
| | II | $7.863 \times 10^{-4}$ | $1.955 \times 10^{-10}$ | $2.010 \times 10^{-1}$ | $2.011 \times 10^{-1}$ |
| | III | $0.000$ | $0.000$ | $4.393 \times 10^{-1}$ | $4.393 \times 10^{-1}$ |
| 10 | I | $9.999 \times 10^{-1}$ | $2.696 \times 10^{-1}$ | $9.667 \times 10^{-1}$ | $1.000$ |
| | II | $2.001 \times 10^{-2}$ | $1.245 \times 10^{-3}$ | $4.240 \times 10^{-1}$ | $4.274 \times 10^{-1}$ |
| | III | $4.775 \times 10^{-12}$ | $1.104 \times 10^{-7}$ | $3.237 \times 10^{-2}$ | $3.237 \times 10^{-2}$ |
| 11 | I | $9.832 \times 10^{-1}$ | $4.237 \times 10^{-1}$ | $9.188 \times 10^{-1}$ | $9.908 \times 10^{-1}$ |
| | II | $1.716 \times 10^{-1}$ | $3.765 \times 10^{-2}$ | $5.704 \times 10^{-1}$ | $5.885 \times 10^{-1}$ |
| | III | $2.408 \times 10^{-4}$ | $6.932 \times 10^{-4}$ | $2.064 \times 10^{-1}$ | $2.065 \times 10^{-1}$ |
| 12 | I | $9.540 \times 10^{-1}$ | $4.555 \times 10^{-1}$ | $8.930 \times 10^{-1}$ | $9.740 \times 10^{-1}$ |
| | II | $2.354 \times 10^{-1}$ | $8.504 \times 10^{-2}$ | $5.613 \times 10^{-1}$ | $5.893 \times 10^{-1}$ |
| | III | $3.342 \times 10^{-3}$ | $6.223 \times 10^{-3}$ | $2.312 \times 10^{-1}$ | $2.324 \times 10^{-1}$ |
| 13 | I | $9.113 \times 10^{-1}$ | $4.746 \times 10^{-1}$ | $8.747 \times 10^{-1}$ | $9.468 \times 10^{-1}$ |
| | II | $2.897 \times 10^{-1}$ | $1.402 \times 10^{-1}$ | $5.570 \times 10^{-1}$ | $5.882 \times 10^{-1}$ |
| | III | $1.662 \times 10^{-2}$ | $2.339 \times 10^{-2}$ | $2.489 \times 10^{-1}$ | $2.534 \times 10^{-1}$ |
| 14 | I | $8.905 \times 10^{-1}$ | $4.807 \times 10^{-1}$ | $8.701 \times 10^{-1}$ | $9.308 \times 10^{-1}$ |
| | II | $3.097 \times 10^{-1}$ | $1.659 \times 10^{-1}$ | $5.578 \times 10^{-1}$ | $5.850 \times 10^{-1}$ |
| | III | $2.713 \times 10^{-2}$ | $3.632 \times 10^{-2}$ | $2.557 \times 10^{-1}$ | $2.620 \times 10^{-1}$ |
| MCS [a] | I | $9.965 \times 10^{-1}$ | $3.931 \times 10^{-1}$ | $9.499 \times 10^{-1}$ | $9.970 \times 10^{-1}$ |
| | II | $7.823 \times 10^{-2}$ | $1.752 \times 10^{-2}$ | $5.667 \times 10^{-1}$ | $5.747 \times 10^{-1}$ |
| | III | $0.000$ | $1.000 \times 10^{-3}$ | $1.580 \times 10^{-1}$ | $1.590 \times 10^{-1}$ |
| Error (%) [b] | I | $0.19$ | $0.3$ | $0.33$ | $0.21$ |
| | II | $3.03$ | $3.05$ | $0.27$ | $0.17$ |
| | III | - | $90$ | $0.53$ | $1.16$ |

[a] The aim of this case is to verify the accuracy of failure probabilities obtained by probabilistic density functions (PDF). [b] The error of $p_f$ between probabilistic density functions and Monte Carlo simulations (Take case 1 as an example).

From Table 4, the failure probability is sensitive to design levels. For the first design level, the failure probability is close to 1, while for the third design level, the failure probability is close to 0. In addition, the influence of $COV\varphi$ on failure probability is less obvious than $COVc_u$ and $\delta_y$. Based on the statistical characters of responses to excavation, similar conclusions are reached in Section 3. For the same level of $COVc_u$, in design level III, system failure probabilities have a tendency to grow with $\delta_y$ when $\delta_y$ ranges from 1~25 m. For design level I, there is a negative correlation between $p_{f_1}$, $p_{f_3}$ and $\delta_y$, while the correlation between $p_{f_2}$ and $\delta_y$ is positive. For design level II, there is a positive correlation between $p_{f_1}$, $p_{f2}$ and $\delta_y$, while $p_{f_3}$ become larger then smaller with the increase of $\delta_y$. The reason is that $\delta_y$ has an impact on the mean value of excavation responses. Therefore, sometimes, the correlation between $\delta_y$ and $p_f$ is not monotonic.

On the other hand, system failure probability $p_f^{sys}$ is larger than single failure probability and it is usually close to the maximum single failure probability. Moreover, if all failure modes are considered independent events, the system failure probability equals the product of each single failure probability. For example, in Case 11 (design level II), $p_f^{sys}$ will be 0.343 if all failure modes are considered independent events. However, in Table 4, $p_f^{sys}$ equals 0.589. Therefore, it is necessary to consider the correlation between failure modes.

As mentioned in Section 3.1, the probability density functions (PDF) of excavation responses were obtained via the K-S test. In order to verify the accuracy of failure probabilities obtained by PDF, case 1 was selected as an example and a 2000-times Monte Carlo simulation (MCS) was conducted.

As shown in Table 4, the errors between the two methods were small except $p_{f_1}$ and $p_{f_2}$ in Design level III. The reason is the limited time of simulations. Both $p_{f_1}$ and $p_{f_2}$ in Design level III is too small, which is lower than $10^{-3}$. In general, the number of MCS should be larger than $10/p_f$ [41,46]. Therefore, the 2000-times MCS cannot verify the accuracy of $p_{f_1}$ and $p_{f_2}$ in design level III. However, the results prove that the probability density function is an efficient method to calculate $p_f$.

## 5. Discussion

Based on the numerical simulation results. Several issues were priceable to further analyze.

(1) For the Cholesky decomposition method, it was quicker and more accurate to generate random field samples compared to K-L expansion and local averaging. However, it is hard for Cholesky to generate random field samples with high correlation variables. For high correlation variables, coupling the variables with high correlation is an effective method.

(2) It is noticeable that the function *integral3* was adopted to calculate the triple integral. If the number of random variables is more than three, the function *integral3* will not be available. However, the function *mvncdf* is available for more than three random variables.

(3) With considering spatial variability of soil parameters, different failure modes will occur during deep excavation, such as wall deformation dominating and basal heave dominating. Different from the deterministic model, the deformation will be subject to elements with lower stiffness in the random finite element model [8]. The dominate deformation depends on the location of the lower stiffness elements. However, for geotechnical engineering, the real distribution of element stiffness is unknown. It is necessary to quantify the effect of uncertainty of soil parameters via reliability analysis.

(4) From Table 4 and Figure 7, friction angle had little effect on failure probability and distribution of deformation responses. Similar conclusions were drawn from probabilistic analysis of vault settlement of tunnel [47]. Therefore, if the deformation index was chosen to evaluate failure probability, the spatial variability of elastic modulus should be considered first.

(5) For system reliability analysis, a key point is how to characterize the correlation between each failure mode. From Sections 3.3 and 4, it is concluded that there is a certain

correlation between failure modes. Due to the limitations of the Pearson correlation matrix and Nataf transformation, the multiple-dimensional log-normal distribution model is used to calculate system reliability, which is a convenient method.

(6)    In this study, it is verified that the spatial variability of soil has a great influence on excavation deformations. For practical engineering, it is necessary to decrease the negative influence. However, the information from the geological survey is not taken to good use. Conditional random field model is an efficient way to integrate the geological data into a random finite element model. How to develop a conditional random finite method and establish a comprehensive system reliability model will be significant for the next studies.

## 6. Conclusions

In this study, a comprehensive investigation of the responses and failure analysis of excavation in spatial variability soil was conducted. A multiple-dimensional log-normal distribution model is adopted to evaluate system reliability, which can take the correlation between multiple failure modes into account. Based on the simulation results, the following conclusions are drawn:

(1)    The spatial variability of soil parameters has a negative effect on the safety of deep excavation. Basal heave is more subjected to soil spatial variability than wall deflection and ground settlement. The distribution of deformations induced by excavation becomes wider with the growth of the scale of fluctuation.

(2)    Different from ultimate limit state analysis, the deformation responses induced by excavations are more sensitive to elastic modulus and soil cohesion than friction angle. In addition, the high uncertainty of soil properties would entail different failure modes such as wall deflection dominating and basal heave dominating.

(3)    The responses of excavation, such as the deflection of the wall, ground settlement, and basal heave, basically follow log-normal distribution via KS-test and SW-test. The fitted probabilistic density functions can be used to carry out reliability analysis, which is an efficient method. Via Latin hypercube sampling technique and K-S test, fewer samples are needed to estimate failure probabilities than in Mote Carlo simulations.

(4)    The multiple lognormal probabilistic density function is a convenient method to describe the correlation between failure modes and calculate the system reliability of deep excavation. System failure probability is usually lower than single failure probabilities. The system failure probability is sensitive to the design level of excavation. It is necessary to determine the design level based on the geometry size of excavation, geological conditions, and surrounding building environment.

**Author Contributions:** Conceptualization, L.H. and X.W.; methodology, L.H.; software, L.H.; validation, L.H., L.C. and X.W.; formal analysis, L.H.; investigation, L.C.; resources, X.W.; data curation, L.C.; writing—original draft preparation, L.H.; writing—review and editing, X.W.; visualization, X.W.; supervision, X.W.; funding acquisition, X.W. All authors have read and agreed to the published version of the manuscript.

**Funding:** This research received no external funding.

**Data Availability Statement:** All data, models, or codes that support the findings of this study are available from the corresponding author upon reasonable request.

**Conflicts of Interest:** The authors declare no conflict of interest.

## Appendix A

**Table A1.** Statistical character of responses of excavation.

| Case Number | Wall Deflection (mm) | | Ground Settlement (mm) | | Basal Heave (mm) | |
|---|---|---|---|---|---|---|
| | $\mu$ | COV | $\mu$ | COV | $\mu$ | COV |
| 1 | 74.91 | 4.778% | 39.38 | 6.644% | 77.95 | 15.657% |
| 2 | 74.94 | 4.858% | 39.30 | 5.559% | 77.97 | 15.308% |
| 3 | 74.97 | 5.068% | 39.32 | 5.618% | 78.18 | 15.258% |
| 4 | 74.24 | 3.249% | 38.78 | 3.735% | 74.02 | 10.157% |
| 5 | 74.28 | 3.446% | 38.79 | 3.757% | 74.15 | 10.132% |
| 6 | 74.36 | 3.702% | 38.83 | 3.880% | 74.39 | 10.119% |
| 7 | 73.86 | 1.838% | 38.49 | 1.933% | 71.63 | 5.026% |
| 8 | 73.92 | 2.139% | 38.50 | 2.148% | 71.78 | 5.031% |
| 9 | 73.99 | 2.481% | 38.54 | 2.477% | 71.98 | 5.048% |
| 10 | 74.79 | 3.434% | 39.12 | 3.767% | 77.51 | 11.188% |
| 11 | 75.21 | 6.789% | 39.54 | 7.437% | 78.77 | 18.279% |
| 12 | 75.45 | 8.697% | 39.76 | 9.355% | 79.05 | 20.696% |
| 13 | 75.76 | 11.001% | 39.97 | 11.564% | 79.38 | 22.888% |
| 14 | 75.91 | 12.127% | 40.08 | 12.734% | 79.57 | 23.388% |

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
