# Peer review of "Reliability Analysis of Serviceability Limit State for Braced Excavation Considering Multiple Failure Modes in Spatially Variable Soil"

_buildings, doi:10.3390/buildings12060722_

Round 1

Reviewer 1 Report

After reading the manuscript “Reliability analysis of serviceability limit state for braced excavation considering multiple failure modes in spatially variable soil”  my comments and concerns are below;

  1. In the current version, the novelty is lacking.
  2. Construction steps are given in the Table. Are these practical; 4-meter excavation and then installation of strut?
  3. The paper is general or a case study? How the authors selected the input parameters? Discuss in details.
  4. Line 119-120; Error! Reference source not found.????
  5. Model validation is missing.
  6. Table numbering needs modification.
  7. Results need comprehensive discussion, specially Table 3.
  8. Conclusion must be based on this study results.
  9. Follow the journal guidelines in the manuscript, especially for references.
  10. Revise the abstract, which must include the introduction, research novelty, methodology and results.
  11. References are very old. Focus on the latest references.

Author Response

Dear Editors and Reviewers:

Thanks for the comments concerning our manuscript entitled “Reliability analysis of serviceability limit state for braced excavation considering multiple failure modes in spatially variable soil” (ID: Buildings-1713133). Those comments are all valuable and helpful for us in revising and improving the manuscript, as well as the important suggestion to the presented study. We have made substantial amendments based on the comments and suggestions. All revisions to the manuscript were marked up using the “Track Changes” function by MS word. In addition, we prepared a MS word file for editors and reviewers. In this PDF file, revised portions except paper formats (such as figures, tables, equations and references) are marked in red. We hope this file can display the important modifications well.

Revision 1: All Tables and figures were modified according to publishing guidelines.

Revision 2: All references were modified according to publishing guidelines. All citations were marked with number. Add the citations of latest studies on random analysis of braced excavation.

Revision 3: Re-summary the conclusions so that the connection between results analysis, discussion and conclusions was closer. Modify abstract and ensure it could summary introduction, novelty, methods and results in this study.

Revision 4: add the verification of numerical models in deterministic analysis.

Revision 5: Made a supplementary explanation according reviewer’s comments, which were marked in red in MS word files.

Point-by-point responses to Reviewers' comments

Comment (1): In the current version, the novelty is lacking.

Responses to comment (1): This study focused on system reliability analysis for deep excavation. This study investigated the effect of spatial variable soil layers on deep excavation and proposed fitted probability density functions method to calculate system reliability. Compared with Monte Carlo simulations and analytic method, proposed method has a higher efficiency. In addition, this study analyzed the effect of multiple failure modes. In order to emphasize our novelty in this study. We modify abstract and conclusions. More details were shown in Responses to Comment (8) and Comment (10).

Comment (2): Construction steps are given in the Table. Are these practical; 4-meter excavation and then installation of strut?

Responses to comment (2): Thanks for your comments. For most excavations, the installation of strut was before excavation. In this study, because the number of numerical models was large, in order to simply the calculation stages, we combine the installation of strut and excavation into one step. We modify Table 2 so that the illustration of construction stages reflects our actual numerical models.

Comment (3): The paper is general or a case study? How the authors selected the input parameters? Discuss in details.

Responses to comment (3): This paper is a general study and aims to investigate the effect of spatial variability of clay soil properties on deep excavation. The numerical case was referred previous studies of Goh et al [1]. The input parameters of clay soil (γ, ν, E, c, φ) were referred previous studies of Goh et al. [1,2,3]. The input parameters of san soil in HS model were referred Zhang et al. [3]. The proportional relationship between of ,  and  equals to , Which is the suggestion of PLAXIS manual [4]. The value of m ranges from 0 to 1, which decreases with the growth of soil stiffness [4]. Both these citations were added into revised paper.

[1] Goh, A.T.C., Zhang, W.G. and Wong, K.S. 2019. “Deterministic and reliability analysis of basal heave stability for excavation in spatial variable soils.” Computers and Geotechnics. 108: 152–160. http://doi.org/10.1016/j.compgeo.2018.12.015.

[2] Goh, A.T.C. 2017. “Deterministic and reliability assessment of basal heave stability for braced excavations with jet grout base slab.” Engineering Geology. 218: 63–69. http://doi.org/10.1016/j.enggeo.2016.12.017

[3] Zhang, W.G., Hong, L. Y.Q., Li, Zhang, R.H., Goh, A.T.C. and Liu., H.L. 2021. “Effects of jet grouting slabs on responses for deep braced excavations.” Underground Space. 6: 185-194. http://doi.org/10.1016/j.undsp.2020.02.002

[4] PLAXIS 2D Manuals. 2018. PLAXIS company: Amsterdam, Netherlands.

Comment (4): Line 119-120; Error! Reference source not found.????

Responses to comment (4): Thanks for your comments. The error in line 275 of original manuscript were corrected. The sentence is corrected as “From Figure 9”. The format of Equations, Figures, Tables and References were modified in detail and checked carefully.

Comment (5): Model validation is missing.

Responses to comment (5): The numerical models and input parameters in this study were referred previous studies. We added the verification of deterministic model in section 2. Based on previous studies on characteristic deformation induced by excavation, the ground settlement curve and wall deflection curve also verified the accuracy of numerical models.

Comment (6): Table numbering needs modification.

Responses to comment (6): Thanks for your comments. All tables and figures were listed in order and citied correctly. There are some problems between transformation from word to PDF in submission system. We will check the number of Tables and Figures in revised paper carefully so that the PDF files generated by system was corrected.

Comment (7): Results need comprehensive discussion, specially Table 3.

Responses to comment (7): In original paper, line 360-387 was the analysis and discussion of table 3, including the effect of scale of fluctuation, the effect of uncertainty of soil properties, the connection between single failure probability and system failure probability. Finally, we verify the results obtained by fitted probability density functions via Monte Carlo method. In addition, we added detailed discussions.

Comment (8): Conclusion must be based on this study results.

Responses to comment (8): In this paper, there are four conclusions:

The first point emphasized the uncertainty of soil properties had a great influence on excavation responses. The standard deviation of excavation responses increases with the growth of COV of soil parameters. The first point was drawn from Figure 5c, 5d, 5e and section 3.1.

The second point focused on the comparison between effect of  and . The second point was drawn from Figure 6 and section 3.1. The discussion of Cholesky decomposition was added into section 5.

The third point summarized the statistical characters of excavation responses. The third point was also drawn from section 3.1 and Figure 5a, 5b.

The fourth point focused on the effect of soil properties uncertainty on system reliability of deep excavation. The fourth point was drawn from section 4.

  In the revised paper, we combined conclusion 1, conclusion 2 and conclusion 3 because they were also drawn from section 3.1. In addition, we re-write the second conclusion to summarize section 3.2 and section 3.3. The new conclusions were shown as following:

In this study, a comprehensive investigation of the responses and failure analysis of excavation in spatial variability soil was conducted. A multiple-dimensional log-normal distribution model is adopted to evaluate system reliability, which can take the correlation between multiple failure mode into account. Based on the simulation results, the following conclusions are drawn:

(1)   The confidence interval of excavation responses become wider with the growth of uncertainty of soft clays. Basal heave is more subjected to soil spatial variability than wall deflection and ground settlement. The excavation responses are more sensitive to elastic modulus and soil cohesion than friction angle. In addition, the high uncertainty of soil properties would entail different failure modes such as wall deflection dominating and basal heave dominating.

(2)   The standard deviation of deformations increases first then keeps stale when scale of fluctuation grows and excavation depth is constant. However, the scale of fluctuation has little influence on failure probability of deep excavation.

(3)   The responses of excavation such as deflection of wall, ground settlement and basal heave basically follow log-normal distribution via KS-test and SW-test. The fitted probabilistic density functions can be used to carry out reliability analysis, which is another efficient method. Via Latin hypercube sampling technique and K-S test, less samples are needed to estimate failure probabilities than Mote Carlo simulations.

(4)   The correlation between deflection of wall, ground settlement and basal heave should be considered in system reliability analysis, otherwise, the system failure probability will be over-estimated or under-estimated. The system failure probability is sensitive to design level of excavation. It is necessary to determine design level based on the geometry size of excavation, geological conditions and surrounding building environment.

Comment (9): Follow the journal guidelines in the manuscript, especially for references.

Responses to comment (9): All references were modified according to publishing guidelines. All citations were marked with number.

Comment (10): Revise the abstract, which must include the introduction, research novelty, Comment methodology and results.

Responses to comment (10): We modify the original abstract and add the summary of background and introduction. The new abstract was shown as following:

High uncertainty is an inherent behavior of geotechnical materials. Nowadays, random field theory an advanced method to quantify effect of high uncertainty on geotechnical engineering. This study investigates the effect of spatial variable soil layers on deformations of deep excavation via random finite element method. A procedure based on PLAXIS 2D software was developed to generate two-dimension random finite element models including multiple variables. Via K-S test and S-W test, the excavation deformations are basically followed lognormal distribution. With the growth of standard deviation of soil properties parameters, the distribution of excavation deformations become wider and the failure probability increases. When vertical scale of fluctuation ranges from 1m to 25m, the distribution of excavation deformations become wider. To analyze system reliability, this study proposed a fitted multiple lognormal distribution method, which was a method with higher efficiency. The results indicated that system reliability was lower than single failure probability and sensitive to design level. The system failure probability will be over-evaluated or under-evaluated if the correlation between excavation responses is ignored. This study provided a novel method to quantify the effect of high uncertainty of soil layer on deep excavation, which is meaningful for excavation reliability design.

Comment (11): References are very old. Focus on the latest references.

Responses to comment (11): Actually, most old references appeared in the introduction. In this part, we introduce the development of random field theory, therefore, some classical studies were mentioned, such as studies of Phoon et al., Vanmarcke et al. and Fenton et al. On the other hand, there is few studies on random analysis for deep excavation, therefore, we introduced the random analysis of slope stability, tunnel reliability and others. The random finite element method was first applied in reliability analysis of shadow foundations and slope stability. In the revised paper, we added the latest studies on reliability analysis for deep excavation.

Reviewer 2 Report

The paper discusses the effect of spatial variability of soil on excavation responses and system reliability using random finite element models in PLAXIS 2D. The work has a numerical nature. Presented test procedure is logical and consistent. I have not objections to this. In my opinion, this paper is interesting. However, Dear Authors, some modifications to the manuscript are necessary before it can be accepted for publication. Particular attention is needed at the following:

ALL MANUSCRIPT - The text editing is very negligent. You should read “Instructions for Authors”.

SPECIFIC COMMENTS:

  1. Please format the Equations according to publishing guidelines.
  2. Please format the Figures according to publishing guidelines. For example, it should be in the text: “Figure 1 shows (…)” instead of “Fig. 1 shows (…)”, etc.
  3. Throughout the manuscript there are the illegible references to Figures, Tables, and Literature. The following text appears: Error! Reference source not found. It is impossible to verify correctness.
  4. Figure 1:
    Please write the soil symbols on the model.
  5. Page 4, Table 1 (and other Tables):
    Incorrect numbering of Tables. Table 1 appears 2 times. Please take into account the change of numbering and references to the Tables throughout the paper.
  6. Page 4, (second) Table 1:
    - You should standardize the notation and consistently give the parameter name, its symbol and unit.
    - Where are the parameter values from? Do they come from own research or from literature?
  7. Page 4, (second) Table 1, and line 135:
    Please standardize the cohesion symbol: either cu or c.
  8. Page 4, lines 135-136:
    For clay soil, elastic modulus is hard to determine via laboratory test.” - Where does this opinion come from? Please explain it. The method of determining Young's modulus is the same for clays and sands. The only difference may be in the duration of the test (and not in the more difficult procedure), which is longer for the clays.
  9. Page 5, line 150:
    It should be: “The sample model in this study is shown in Figure 2.” instead of “The sample model in this study is shown in Fig. 1.”.
  10. Page 6, line 171:
    “In this paper, there are 13 cases shown in Table 3” - Shouldn't it be: “In this paper, there are 14 cases shown in Table 3”?
  11. Page 6, Table 2:
    Incorrect table numbering.
  12. Figures 4, 5 and 7:
    The poorly visible inscriptions on the drawing (and on the axes). Please enlarge the font.
  13. Page 8, Figure 5:
    Please sign which drawing is a), b), c), etc.
  14. Page 8, Figure 5 and Page 9, Figure 5:
    Incorrect numbering of Figures. Figure 5 appears 2 times. Please take into account the change of numbering and references to the Figures throughout the paper.
  15. Page 10, Figure 6:
    Please sign the drawing correctly. There should be (a) and (b).
  16. Page 12, Equation 5:
    Lack of the explanation of what epsilon1,2,3 means?
  17. Page 12, Lines 295-296:
    “JGJ 295 120-2012 (…) GB 50007-2011” - missing in References
  18. References:
    Please format the References according to publishing guidelines.

Author Response

Dear Editors and Reviewers:

Thanks for the comments concerning our manuscript entitled “Reliability analysis of serviceability limit state for braced excavation considering multiple failure modes in spatially variable soil” (ID: Buildings-1713133). Those comments are all valuable and helpful for us in revising and improving the manuscript, as well as the important suggestion to the presented study. We have made substantial amendments based on the comments and suggestions. All revisions to the manuscript were marked up using the “Track Changes” function by MS word. In addition, we prepared a MS Word file for editors and reviewers. In this PDF file, revised portions except paper formats (such as figures, tables, equations and references) are marked in red. We hope this MS Word file can display the important modifications well.

Revision 1: All Equations were modified according to publishing guidelines. Each equation was put into a single-row table without frame.

Revision 2: All references were modified according to publishing guidelines. All citations were marked with number.

Revision 3: All figures were modified according to publishing guidelines. All “Fig” were corrected as “Figure”. The subtitles of figures were corrected as (a), (b), (c) etc. Modified the figures by following reviewer’s comments.

Revision 4: Made a supplementary explanation according reviewer’s comments, which were marked in red in PDF files.

Point-by-point responses to Reviewers' comments

Comment (1): Please format the Equations according to publishing guidelines.

Responses to comment (1): All Equations were modified according to publishing guidelines. Each equation was put into a single-row table without frame.

Comment (2): Please format the Figures according to publishing guidelines. For example, it should be in the text: “Figure 1 shows (…)” instead of “Fig. 1 shows (…)”, etc.

Responses to comment (2): All figures were modified according to publishing guidelines. All “Fig” were corrected as “Figure”.

Comment (3): Throughout the manuscript there are the illegible references to Figures, Tables, and Literature. The following text appears: Error! Reference source not found. It is impossible to verify correctness.

Responses to comment (3): Thanks for your comments. The error in line 275 of original manuscript were corrected. The correct expression of this sentence is “From Figure 9”. The format of Equations, Figures, Tables and References were modified in detail and checked carefully.

Comment (4):

Figure 1:

Please write the soil symbols on the model.

Responses to comment (4): The name of soil models and retaining structure models were added into Figure 1.  The soil models and structure models were consisting with Table 1 and Table 2.

Comment (5):

Page 4, Table 1 (and other Tables):

Incorrect numbering of Tables. Table 1 appears 2 times. Please take into account the change of numbering and references to the Tables throughout the paper.

Responses to comment (5): Thanks for your comments. All tables and figures were listed in order and citied correctly. There are some problems between transformation from word to PDF in submission system, similar problem appears in comment (11), comment (12) and comment (14). We will check the number of Tables and Figures in revised paper so that the PDF files generated by system was corrected.

Comment (6):

Page 4, (second) Table 1:

- You should standardize the notation and consistently give the parameter name, its symbol and unit.

- Where are the parameter values from? Do they come from own research or from literature?

Responses to comment (6): In the revised paper, it was corrected as Table 2. We standardize the parameter name, symbol and unit. The value of parameters was referred from both literatures and our previous studies, the related references were also listed. In addition, we added the explanation of symbol in Table 2.

Comment (7):

Page 4, (second) Table 1, and line 135:

Please standardize the cohesion symbol: either cu or c.

Responses to comment (7): We check the paper carefully and standardize the cohesion model. The cohesion was represented by cu.

Comment (8):

Page 4, lines 135-136:

“For clay soil, elastic modulus is hard to determine via laboratory test.” - Where does this opinion come from? Please explain it. The method of determining Young's modulus is the same for clays and sands. The only difference may be in the duration of the test (and not in the more difficult procedure), which is longer for the clays.

Responses to comment (8): Thanks for your comments. Generally, the geological survey report usually supplied compressive modulus of soil. For most engineering, we obtained compressive modulus rather than elastic modulus. However, in finite element model, elastic modulus was necessary. Based on the previous studies, it is a convenient way to determine elastic modulus of clay soil via empirical method. The related studies were also citied. We modify this part so that the expression will be accurate and serious.

Comment (9):

Page 5, line 150:

It should be: “The sample model in this study is shown in Figure 2.” instead of “The sample model in this study is shown in Fig. 1.”.

Responses to comment (9): In this sentence, the sample model means a general random finite element model. Different from deterministic model, in random model, the value of soil properties depends on location in space. However, there is no difference between the geometry model, boundary conditions and construction steps in random model and deterministic model. Figure 2 is the whole framework of random finite element method; the first steps aims to set up a general random finite model. The expression of first steps was modified as: “Establish a sample model of random finite element method. Different from deterministic model shown in Figure 1, the clay layer was divided into 560 regions, corresponding to random field elements. Meanwhile, the statistical characters of random fields are determined, such as the mean value (μ), Coefficient of variation (COV), correlation matrix R and fluctuation of scale in horizontal and vertical direction (δx and δy).

Comment (10):

Page 6, line 171:

“In this paper, there are 13 cases shown in Table 3” - Shouldn't it be: “In this paper, there are 14 cases shown in Table 3”?

Responses to comment (10): Actually, there are 14 kinds of cases in this paper, we already modify this sentence.

Comment (11):

Page 6, Table 2:

Incorrect table numbering.

Responses to comment (11): The number of all tables were modified and check carefully. In the revised paper, there are totally four tables.

Comment (12):

Figures 4, 5 and 7:

The poorly visible inscriptions on the drawing (and on the axes). Please enlarge the font.

Responses to comment (12): The original font of Figures 4, 5 and 7 is 20. The revised paper, we adjust the font of text in figures into 24.

Comment (13):

Page 8, Figure 5:

Please sign which drawing is a), b), c), etc.

Responses to comment (13): Thanks for your comments. According to publishing guidelines, all subtitles of figures in this paper were corrected as (a), (b) (c) etc.

Comment (14):

Page 8, Figure 5 and Page 9, Figure 5:

Incorrect numbering of Figures. Figure 5 appears 2 times. Please take into account the change of numbering and references to the Figures throughout the paper.

Responses to comment (14): The number of Figures were sorted again. In the revised paper, there are totally 11 groups of Figures.

Comment (15):

Page 10, Figure 6:

Please sign the drawing correctly. There should be (a) and (b).

Responses to comment (15): The subtitle of Figure 6 was corrected as “Figure 7. Effect of spatial variability on failure mode (a) basal heave dominate (b) wall deflection dominate”.

Comment (16):

Page 12, Equation 5:

Lack of the explanation of what epsilon1,2,3 means?

Responses to comment (16): ε1, ε2 and ε3 represent three kinds of failure functions, [δb δs δd] were considered as random variables. For example, when ε1 is lower than 0, it means the retaining system was considered as failure. Therefore, the failure probability can be estimated via an integral, as shown in Eq (6). In addition, we added the explanation and taken place of [ε1 ε2 ε3] with [f(δb) f(δs) f(δd)] so that the symbols were the same in Eq (5) and Eq(6).

Comment (17):

Page 12, Lines 295-296:

“JGJ 295 120-2012 (…) GB 50007-2011” - missing in References

Responses to comment (17): These two references were supplied, which was [35] and [36], respectively.

Comment (18):

References:

Please format the References according to publishing guidelines.

Responses to comment (18): All references were modified according to publishing guidelines. All citations were marked with number. In addition, we add some new references and citations on this topic.

Round 2

Reviewer 1 Report

Plz find the attached comments

Reviewer 2 Report

Dear Authors,
I thank you for taking account of almost all my comments and suggestions. I appreciate your efforts to make your study more attractive for the Reader. However, there are still a few elements to be clarified or supplemented. First of all, please arrange Table 2 and drawings and explain the meaning of the parameters: E(ref/50), E(ref/oed), E(ref/ur).  I have marked all my comments in the attached manuscript copy.

Round 3

Reviewer 1 Report

  1. Check the Tables, Figures, and equation numbering. Also verify their numbering in the text.

Author Response

Dear Editors and Reviewers:

Thanks for the careful guidance for our manuscript entitled “Reliability analysis of serviceability limit state for braced excavation considering multiple failure modes in spatially variable soil” (ID: Buildings-1713133). Those comments are all valuable and helpful for us in revising and improving the manuscript, as well as the important suggestion to the presented study. We have made substantial amendments based on the comments and suggestions. All revisions to the manuscript were marked up using the “Track Changes” function by MS word.

Revision 1: All equations were edited again by mathtype software 6.9 vision.

Revision 2: Figures were put into Tables without frame lines.

Revision 3: Added the references and illustration of why choose MC model according to academic editors.

Point-by-point responses to Reviewers' comments

Comment (1): Check the Tables, Figures, and equation numbering. Also verify their numbering in the text

Responses to Comment (1): Thanks for your comment. We check and modify the Tables, Figures, and equations according to academic editors.

Reviewer 2 Report

In my opinion, the Equations and Figures are not properly formatted, but I leave it to the Chief Editor's decision. Apart from that, I have no other comments.

Author Response

Dear Editors and Reviewers:

Thanks for the careful guidance for our manuscript entitled “Reliability analysis of serviceability limit state for braced excavation considering multiple failure modes in spatially variable soil” (ID: Buildings-1713133). Those comments are all valuable and helpful for us in revising and improving the manuscript, as well as the important suggestion to the presented study. We have made substantial amendments based on the comments and suggestions. All revisions to the manuscript were marked up using the “Track Changes” function by MS word.

Revision 1: All equations were edited again by mathtype software 6.9 vision.

Revision 2: Figures were put into Tables without frame lines.

Revision 3: Added the references and illustration of why choose MC model according to academic editors.

Point-by-point responses to Reviewers' comments

Comment (1): In my opinion, the Equations and Figures are not properly formatted, but I leave it to the Chief Editor's decision. Apart from that, I have no other comments.

Responses to Comment (1): Thanks for your comment. We check and modify the Tables, Figures, and equations according to academic editors.